# Development and Testing of Castables with Low Content of Calcium Oxide

**DOI:** 10.3390/ma15175918

**Published:** 2022-08-26

**Authors:** David Zemánek, Lenka Nevřivová

**Affiliations:** Faculty of Civil Engineering, Brno University of Technology, Veveří 331/95, 602 00 Brno, Czech Republic

**Keywords:** sol-gel, refractory, castables, corrosion

## Abstract

Colloidal silica is used in many kinds of industry. It is an aqueous dispersion of SiO_2_ nanoparticles. SiO_2_ colloidal solutions are commercially available in different concentrations, with different particle sizes and are stabilized with different ions. Colloidal SiO_2_ was used in this study as a cement replacement in refractory castable. The present study, in its first stage, offers an assessment of five different SiO_2_ colloidal solutions. The particle size of the solutions was 15 nm, the particle concentration was 30% and 40% and the colloidal solutions were stabilized with Na^+^, OH^−^ and Cl^−^ ions. The effect of the colloidal solutions on selected characteristics of the refractory pastes and on their mineralogical composition after firing at 1000 °C and 1500 °C was described. The most suitable SiO_2_ colloidal solution from the first stage was subsequently used for the refractory castable test samples’ preparation in the second stage. Refractory castables, unlike paste, contain a coarse aggregate (grog) up to a grain size of 6 mm. Four types of coarse refractory grog were evaluated. Their effect on selected characteristics of the refractory castable and on its mineralogical composition after firing at 1000 °C and 1500 °C was described. The selected characteristics, within the scope of this study, include bulk density, apparent porosity, cold modulus of rupture and linear changes after firing. Finally, the study describes the effect of the sol particle concentration and the effect of pore size distribution on corrosion resistance and on the internal structure of the material. Mineral and chemical compositions and microstructures of both the raw materials and designed aggregates were thoroughly investigated by the means of X-ray fluorescence spectroscopy, powder X-ray diffraction and scanning electron microscopy. An analysis of the transition zone between corrosive media (K_2_CO_3_) and tested castables showed better corrosion resistance for a sol-gel castable than an ultra-low cement castable.

## 1. Introduction

There has been worldwide growth of castable usage, mainly attributed to the fact that more and more brick-laid linings have been and can further be replaced by castables that possess a variety of advantages over bricks, in terms of production cost, installation efficiency, safety, material consumption, etc. Monolithic ceramics have evolved over the years into a widely used class of refractories. The progress of monolithic refractories is closely related to improvements in the quality of raw materials, new binders, ultra-fine powders, and efficient additives [1]. Refractory castables can be classified based on different aspects including the calcium oxide content, binder source, chemical composition, bulk density, application method and others [2]. A medium cement castable (MCC) is characterized by a calcium oxide content higher than 2.5%, a low cement castable (LCC)’s CaO content is 1.0–2.5%, an ultra-low cement castable (ULCC)’s CaO content is 0.2–1.0% and a no cement castable (NCC) CaO content until 0.2% [3]. Different sorts of binding systems have been developed throughout the years, starting with hydraulic bonding, in which higher amounts of calcium aluminate (CA) cement were used, towards coagulating binders such as colloidal silica or alumina [4].

Hydraulic binders were used at the very beginning of castable production. Conventional cement-bonded castables were produced since the early 1920s until the 1960s when CA cement started to be used. The purity of CA cement was rising over the time and in the 1990s, the next step of evolution was done. Hydratable alumina (HA) has been used as a castable binder. It is a transition alumina, having specific crystallinity containing over 90% of alumina. It forms Al_2_O_3_·3H_2_O (AH_3_) and Al_2_O_3_·H_2_O (AH) gel in combination with water which, upon heating, dehydrate and eventually form ceramic bonds [5]. Lower porosities are achieved in the samples as a consequence of the ultrafine pore structure resulting from hydratable alumina. The HA-bonded products thus have an improved corrosion resistance to liquids and gases. Compared to the cement-bonded system, HA requires a longer mixing time and the requirement for water is also high due to the high specific surface area of the binder [6,7]. Thus, they have a specific application to areas where the cement removal is beneficial and where drying out can be controlled tightly, such as in precast [8]. The very new type of hydraulic binders is spinel-containing CA cement and calcium magnesia alumina (CMA) cement [9]. The production of chemically bonded castables started in the 1950s with the usage of phosphates and water glass. Sulfate- and chloride-bonded castables appeared a bit later. Chemical binders promote bonding by the formation of new products or by a polymerization reaction between the binder and the refractory material (oxide aggregates) or between the binder and the hardener [4,10].

Coagulation bonding began to be used in the 1980s. The binders which work by coagulation binding are fine clay powder, ultrafine oxide powder, silica sol, alumina sol, etc. Nanoparticles of the colloidal suspension are attracted by Van der Waal forces, if the equilibrium in the sol is disturbed, it leads to the coagulation of these particles. The addition of a coagulant increases the coagulation activity. When colloidal silica (silica sol) is mixed with a refractory micro filler, the equilibrium of the sol is disturbed and a gel forms around the microparticles which holds the particles together. The gel is responsible for the manipulation strength in the green state [10].

Among the available colloidal systems, only silica sol is commercially exploited as a binder for refractory systems. The high-stability, high-solid content, as well as the possibility of mullite formation at a low temperature in high alumina systems, favors its wide industrial application. However, the presence of free silica in the system can promote the appearance of a vitreous phase and thus decrease refractoriness, which is especially critical in alkali-containing systems, restricting its use for high-temperature applications [11]. Carbon bonding (MgO-C) and nano-engineered castables started to be used in the beginning of the 21st century. Nano engineered castables use colloidal binders and sintering additives at one time [12].

The advancements in nanotechnology in the last two decades could benefit the refractory industry if explored properly [13,14]. The literature shows that the utilization of nanopowders and colloidal suspensions in refractory castables has increased in recent years, mainly to improve the bonding and densification in castables at lower sintering temperatures [13,15,16]. The agglomeration of nanopowder particles (due to its high reactivity owing to a high surface area) results in poor dispersion in the matrix and thus possesses a challenge to use these as additives [17,18]. If the agglomeration problem is kept under control, the use of a nanoparticle addition could improve the castable properties. Another limitation which hinders the use of nanopowders as additives is the availability and the cost of available nanopowders. Nanoparticles containing colloidal suspensions (colloidal binders) are a better alternative and are preferred [18].

A colloid or sol is a stable dispersion of particles in water, wherein particles are too small for gravity to make them settle. Particle size typically ranges from 1 to 1000 nanometers [19]. The name sol-gel derives from the fact that microparticles or molecules in a solution (sols) agglomerate which, under controlled conditions, eventually link together to form a coherent network (gel) [20]. The various advantages associated with sol-gel-bonded castables are less mixing time, cohesive and self-flowing nature, easier installation as there are no vibration works, shorter drying time and reduced drying effects due to a lower moisture content, better corrosion resistance (absence of CaO), better oxidation resistance, better refractoriness, longer lining life and larger self-life as no hydratable phase is present [21]. The bonding action of colloidal binders is based on the gelation (coagulation) of the colloidal particles, which leads to a high permeable and porous material structure. Technically this can be advantageous as the composition can be safely and quickly dried; reducing the risk of explosive spalling and thus the overall processing time is reduced [13].

Various colloidal systems are available for refractory castables production, e.g., silica sol, alumina sol, mullite sol and spinel sol. The use of alumina sol as a binder for refractory castables is mentioned by a few authors only [15,18,22,23,24,25,26]. The advantage of alumina sol is its purity. The use of mullite sol as a binder for refractory castables is mentioned as well, for example, in combination with CA cement, A cement or micro silica [21,27,28,29]. Use of spinel sol as a binder has also been reported by some authors in ultra-low cement compositions [21,27]. However, silica sol is the most common for the refractory castable production. Recently, several studies were focused on the “cement free” refractory castables [30,31].

That is why silica sol was chosen for the production of high-alumina no-cement castable samples in the present study. The study evaluates the properties of no-cement castable samples bonded by a sol-gel method together with an ultra-low cement castable sample, for comparison. Other possible alternative bonds, which can be suitable for comparison as a clay bond, kaolin bond or hydratable alumina bond, were not assumed. Possible applications of developed castables may be in the glass industry (mechanically stressed parts), primary aluminum industry (anode coke oven) or stressed parts of boilers.

The corrosion of refractories can be defined by the loss of thickness and mass from the exposed face of the refractory as a consequence of a chemical attack by a corroding fluid in a process in which the refractory and the corroding fluid react, approaching the chemical equilibrium in the zone of contact [32,33]. The influence of the used slag and influence of the used refractory grog on the microstructure was observed. There are a few tests used for the corrosion resistance determination of refractory material and they are either static or dynamic, such as hot stage microscopy, the crucible test, finger test, dynamic finger test, induction furnace test or rotary slag test [34]. The crucible test was applied in our study. The corrosion resistance of the sol-gel-bonded castables is assumed to be better than the resistance of the ultra-low cement castable. Spinel and silica sols-bonded castables were found to achieve the best results [35].

## 2. Experimental Procedure

### 2.1. Methods

Chemical composition analysis of the raw materials was performed by wavelength-dispersive X-ray spectroscopy (WDXRF) using SPECTROSCAN MAKC-GV (Spectron Company, St. Petersburg, Russia) instrument equipped with QUANTITATIVE ANALYSIS software (version 4.0, Spectron Company, St. Petersburg, Russia). The samples were analyzed in forms of fused beads. Powder X-ray diffraction analysis of the raw materials and engineered aggregates was conducted on Panalytical Empyrean diffractometer (Panalytical B.V., Almelo, The Netherlands) equipped with Cu-anode, 1-D position-sensitive detector at convention Bragg–Brentano reflection geometry. The setting was: step size—0.013° 2θ, time per step—188 s, and angular range 5–80° 2θ. Quantitative phase analysis was done via the Rietveld method using Panalytical High Score 3 plus software (version 4.8, Panalytical B.V., Almelo, The Netherlands).

Scanning electron microscopy with X-ray microanalysis (SEM/EDS) was conducted on gold-coated mechanically broken specimens (for morphological analyses) and on polished carbon-coated thin sections (for chemical microanalyses) using TESCAN MIRA 3 instrument (Tescan Orsay Holding a.s., Brno, Czech Republic) with the accelerating voltage of 30 kV.

Cold crushing strength (CCS) according to standard EN 993-5:2018 (using machine MEGA 11-600 D-S, Brio Hranice s.r.o., Hranice, Czech Republic) and cold modulus of rupture (CMOR) according to standard EN 993-6:2018 (using machine Testometric M350-20CT, Testometric Co. Ltd., Rochdale, UK) were carried out after drying and after firing. The apparent porosity, water absorption and bulk density were determined by a vacuum water absorption method with subsequent hydrostatic weighing (standard EN 993-1:2018). Determination of permanent change in dimensions on heating was tested according to standard EN 993-10:2018.

Abrasion test was carried out under conditions of standard ASTM C 704. Abrasion test used SiC material with particle size 250 µm with feed rate 50 g·min^−1^ and impact angle 90°.

Corrosion testing was carried out according to CEN/TS 15418:2006. Prepared castables were casted into silicone mold with required parameters of the testing samples—outer dimensions of the samples were 100 × 100 × 100 mm, inner hollow cylindrical shaped crucible was 55 mm in diameter and 55 mm in depth. Used corrosion medium was K_2_CO_3_, specifically 20 g was precisely weighed in each tested crucible before testing. Corrosion cup test or crucible test, as it can be entitled, was carried out at 950 °C to achieve melting point of K_2_CO_3_ in this study. Standard CEN/TS 15418:2006 specifies the heating rate for the corrosion cup test to 5 °K/min and soaking time to five hours at the maximum temperature.

Apparent porosity and pore size distribution were determined by mercury intrusion porosimetry using Thermo Finnigan POROTEC Pascal 140–240 instruments (ThermoFisher Scientific, Waltham, MA, USA) with SOL.I.D software (version 3.0.1, ThermoFisher Scientific, Waltham, MA, USA).

### 2.2. Raw Materials and Mixtures

#### 2.2.1. First Stage—Fine Pastes

Raw materials used in this study were obtained from their producers—andalusite (A) from Imerys (Glomel, France), tabular alumina (TA) from Almatis (Ludwigshafen, Germany), reactive alumina (RA) from Nabaltec (Schwandorf, Germany), ground alumina (GA) from Nabaltec (Schwandorf, Germany) and silica fume (SF) from RW Silicium (Pocking, Germany). Colloidal silicas used in this study were chosen from SChem (Ústí nad Labem, Czech Republic). For comparison with traditional hydraulic bond used for ULCC, calcium aluminate cement (CAC) from Almatis (Ludwigshafen, Germany) was chosen, Chemical and phase compositions of the used raw materials are given in Table 1. As the gelling agent, material responsible for transformation from colloidal sol to gel, ammonium chloride (NH_4_Cl, 1 M) from PENTA (Prague, Czech Republic), was used.

In the first stage, fine matrix was prepared from dry raw materials, which were precisely weighed (Table 2), mixed and well homogenized in a laboratory homogenizer for 1 h. After homogenization process, mixtures were mixed for 15 min with designed amount of colloidal silica 8.5% to achieve fine paste. Then, addition of 0.1% of gelling agent (NH_4_Cl) was carried out with subsequent mixing for 5 min.

Fine pastes designation and the type of colloidal silicas stabilization is shown in Section 3.1. Type of stabilization is declared by its producer. Colloidal silica designation is typically, e.g., K1530—where K is a commercial name, 15 represents average particle size diameter d_50_ and 30 is a concentration of solid particles.

#### 2.2.2. Second Stage—Castables

Four types of refractory grog with different amounts of Al_2_O_3_ were used—fused mullite (M) from MOTIM Electrocorundum (Mosonmagyaróvár, Hungary), fireclay grog (F) from P-D Refractories (Velké Opatovice, Czech Republic), high alumina grog (H) from P-D Refractories (Velké Opatovice, Czech Republic) and andalusite (A) from Imerys (Glomel, France). Their chemical and phase compositions are shown in Table 3.

Castables were designed in accordance with Andreasen’s particle distribution model with packing coefficient q = 0.24. Deflocculants or plasticizers were not used in this study. Designed composition of castables based on mentioned coarse grogs is presented in Table 4. Castables are labeled, e.g., NA—where the first letter indicates the type of castables (N—no-cement castable, U—ultra-low cement castable), the second letter represents the type of used refractory grog (A—andalusite, H—high-alumina grog, F—fireclay, M—fused mullite).

### 2.3. Samples Preparation

#### 2.3.1. First Stage—Fine Pastes

Fine pastes were casted into molds with dimensions 20 × 20 × 100 mm and covered by plastic foil. Samples were demolded after 24 h. Green bodies were then dried in the laboratory dryer at 110 °C for 24 h. All the test samples were then fired in an electrical laboratory furnace with air atmosphere at 1000 °C and 1500 °C (heating rate of 4 °K/min and soaking time of 300 min at the maximum temperature).

In the second stage, castables were prepared from dry raw materials. After dry homogenization process that lasted for 1 h, mixtures were mixed for 5 min with designed amount of water, colloidal silica and calcium aluminate cement (CAC) for UA mixture. For no-cement castables (NCC), gelling agent was added to the mixture and second wet mixing lasted 2 min.

#### 2.3.2. First Stage—Castables

Mixtures were casted into molds with dimensions 40 × 40 × 160 mm and covered by plastic foil. Samples were demolded after 24 h. Green bodies were then dried in the laboratory dryer at 110 °C for 24 h. All the test samples were then fired in an electrical laboratory furnace with air atmosphere at 1000 °C and 1500 °C (heating rate of 4 °K/min and soaking time of 300 min at the maximum temperature).

## 3. Results and Discussion

### 3.1. First Stage—Fine Pastes

The objective of the first stage of this study was to describe the effect of colloidal silica on the properties of the fine pastes after drying and after firing. The raw material used for the preparation of the fine matrix is shown in Table 1. The formula for the preparation of the fine matrix was designed based on previous experience and research [36] (see Table 2). Ammonium chloride (NH_4_Cl, 1 M) was used as the gelling agent.

The main output of this stage is the determination of the most suitable colloidal SiO_2_ solution for the preparation of the no-cement refractory castable.

Table 5 presents the results of the experiments that utilized the vacuum water absorption method, together with hydrostatic weighing to determine the apparent porosity (AP) and bulk density (BD), as well as the results of the Cold Modulus of Rupture (CMOR) and linear change after firing (LCf) for all five tested fine pastes.

When fired at 1000 °C, the lowest porosity was achieved using colloidal silica K1530AD. When fired at 1500 °C, the apparent porosity decreased by up to 5.5%. When using a colloid with a higher SiO_2_ solids content (40%), the fine paste has a higher porosity after firing at 1500 °C than when using a colloidal silica with a lower solids content (30%). The lowest decrease in porosity was observed when using L1540 (see Figure 1).

For sample B, which is also stabilized with Na^+^ ions but contains 10% more of silica nanoparticles, the apparent porosity after firing at 1500 °C is 2% higher. This also may be observed in terms of linear shrinkage after firing, where the higher content of silica nanoparticles in sample B leads to lower linear shrinkage at both firing temperatures. Using colloidal silica with a higher content of solids determined the lowest decrease in apparent porosity and a low increase in bulk density in comparison to the concentration of solids at 30%. The highest bulk density of 2608 kg∙m^−3^ was measured for sample D after firing at 1500 °C, where the second highest linear shrinkage of −2.2% was measured and the highest CMOR of 6.6 MPa was obtained. Other results at both firing temperatures are comparable and in the case of physical and mechanical properties, the four tested silicas with the same particles size and concentration provided similar results.

The XRD diffractograms of tested pastes after firing at 1000 °C and 1500 °C are presented in Figure 2.

The major crystalline phases for samples fired at 1000 °C are andalusite (Al_2_(SiO_4_)O), corundum (Al_2_O_3_) and quartz (SiO_2_). The major crystalline phases for samples fired at 1500 °C are andalusite, corundum and mullite (3Al_2_O_3_·2SiO_2_). Major differences in the samples’ mineralogical composition are shown in the following Figure 3. Residual quartz is contained only in samples after firing at 1000 °C. The mullite presence was detected at the main mullite peaks with an intensity of 3.39(1), 3.428(0.95) and 2.206(0.6) only after firing at 1500 °C. At that temperature, all quartz was transformed to mullite and the glassy phase, which is also presented in Table 4.

The mineral andalusite was identified after firing at 1000 °C on all major diffraction lines, 5.54(1), 2.77(0.9) and 4.53(0.9). Corundum 2.085(1), 2.552(0.9) and 1.601(0.8) and quartz 3.342(1) were also identified after firing at 1000 °C. After firing at 1500°C, the content of andalusite and corundum was reduced at the expense of mullite and the glassy phase. The decrease in the andalusite content is documented on diffraction lines 25.3, 39.6, 39.9, 41.4, 41.5 and 41.6° 2θ. A decrease in the corundum content is documented on diffraction lines 25.6 and 41.7° 2θ (see Figure 3 and also Table 6).

Based on the discussed results, silica sol with the designation L1540, sample formula B, which contains 40% of solid SiO_2_ particles, was selected. The minimum LSf values (−0.1% at 1000 °C, −1.0% at 1500 °C) were recorded when using this sol and good physical and mechanical properties were obtained. The highest mullite content was measured as 71.3% for Sample B due to the highest content of solid particles (40%). The higher mullite content in the fine paste when using L1540 (B) is also documented in Figure 3 on the diffraction lines 26.0, 26.3, 39.3 and 40.9° 2θ. In terms of used colloidal silicas with 30% of particles, the highest content was measured as 71.1% for Sample A. The higher mullite content in the prepared fine pastes when colloidal silica with a higher solid-phase content was used is a positive signal and precondition to achieve good physical, mechanical and refractory castable properties.

### 3.2. Second Stage—Castables

Based on the results of the first stage, silica sol L1540 was selected and used in further testing. The second stage was focused on the design of the refractory castable with coarse aggregate (grog) fractions of 0–1, 1–3 and 3–6 mm. The NCC bond was realized by the sol-gel method and was tested with refractory grog with different Al_2_O_3_ contents. The effect of the NCC and ULCC bond on the basic physical and mechanical properties of refractory castable was also compared in this stage.

Five types of castables were prepared in the terms mentioned above. Table 7 shows the chemical composition of the tested mixtures after firing, all tested mixtures met the condition of standard ASTM C401-12:2018 for the calcium oxide content for ultra-low cement castables (ULCC) and no-cement castables (NCC).

#### 3.2.1. Physical and Mechanical Properties after Firing

Table 8 presents castables’ physical properties after drying and firing at selected temperatures.

The apparent porosity (shown in Figure 4 and further discussed in Section 3.2.3.) and water absorption decreases with the increasing firing temperature. The less Al_2_O_3_ contained in the castable, the higher the decrease of WA and AP was measured to be. The porosity decreasing is also related to the further sintering of the coarse grog with a lower alumina (F, H) content.

A comparison of the values for the dried state with the maximum firing temperature shows the highest decrease for sample NF, where the water absorption declined by 1.0% and the apparent porosity declined by 2.7%. Differences in the bond type for UA and NA in terms of apparent porosity and water absorption are similar. The used bond affected the apparent porosity and water absorption marginally. The porosity and mineralogical composition of the used coarse grog is crucial for the final porosity of the refractory castable.

Figure 5 represents linear changes that occurred during the firing of tested castables. After firing at 1000 °C, the sample NA (sol-gel bonded) performed a permanent linear change of 0.00% in comparison to UA (CAC bonded), where an expansion of 0.10% was measured.

#### 3.2.2. Abrasion Test

Andalusite (A) is well-known for its expansion behavior during firing and this phenomenon is obvious by the highest expansion rates, as shown in Figure 5. The opposite mechanism was measured for mullite (M), where the highest shrinkage occurred after firing at both tested temperatures. The most dimensionally stable grog appears to be high alumina grog (HA) with miniscule expansion 0.11% at 1500 °C. To achieve a similar LCf value using andalusite, a combination with mullite is necessary to compensate their expansion and shrinkage behavior after firing.

An abrasion test was carried out under conditions of standard ASTM C 704. Figure 6 represents the loss of abrasion tested at room temperature for tested castables after pre-firing at 1500 °C. The abrasion test used SiC material with a particle size 250 µm with a feed rate of 50 g·min^−1^ and impact angle of 90°. Comparing the NCC refractory castable, the use of a fireclay grog F (abrasion = 5.3 cm^3^) appears to be the most suitable and a mullite grog M (abrasion = 6.8 cm^3^) is the least suitable. NH and NA castables performed the same value of abrasion loss at 5.8 cm^3^. For the refractory castable with andalusite grog, the effect of the used bond on the abrasion resistance was investigated. When the sol-gel bond is used (NA), the abrasion resistance is 17% lower than when using the cement bond (UA). It was shown that abrasion was 17% lower using sol-gel bond (NA) than the calcium cement bond (UA), despite the fact that the apparent porosity and average pore diameter of the UA and NA refractory castables were very similar.

#### 3.2.3. Pore Structure

The effect of the used refractory grog on the pore structure of the refractory castables after firing at 1500 °C is shown in Figure 7. The pore size distribution of the refractory castable with mullite (M) and castable with high alumina (H) grog is very similar. The average pore diameter is 17.48 and 13.1 µm, respectively. Ninety percent of the pores are 1–40 µm in diameter. When andalusite (A) is used, the apparent porosity is lower, as is the average pore diameter, which is 1.24 µm and 2.42 µm, respectively. Seventy-six percent of the pores are within the 1–40 µm interval. For the refractory castable with andalusite grog, we can also compare the effect of the bond used in the structure of the refractory castable. When the sol-gel (NA) bond is used, there are more fine pores in the material structure than when the hydraulic (UA) bond is used, with almost the same apparent porosity (14.9% and 14.2%, respectively). The average pore diameter when sol-gel bonding is used is 50% smaller than when using the hydraulic bond.

#### 3.2.4. Corrosion Crucible Test and EDS Element Mapping

The corrosion crucible test was carried out according to the standard CEN/TS 15418:2006. Crucibles were pre-fired at 1500 °C with a soaking time of 5 h at the maximum temperature, with a heating rate of 1.5 °K/min. Crucibles were filled with a corrosive medium (K_2_CO_3_) and subsequently fired at 950 °C with a soaking time of 5 h at the maximum temperature, with a heating rate 1.7 °K/min.

Based on the general state of the specimen after the corrosion test, the standard [37] defines four stages of corrosion. The decisive parameter for assessing the degree of corrosion is the size of the corroded area and the maximum depth of penetration of the sample by the corrosive agent on a cross section of the test specimen (U: unaffected, for samples with no visible degradation; LA: lightly attacked, for samples with a minor attack; A: attacked, for samples with clearly visible degradation; C: corroded, for completely corroded samples).

The next section is focused on specifically UA and NA castables to compare differences between the sol-gel bond and the hydraulic bond from CAC. A visual evaluation performed on cross-cut crucibles and its results are presented in Appendix A, Figure A1 for the NA castable and in Appendix B, Figure A3 for the UA castable. According to the standard, examined castables may be labeled as unaffected (U) for both samples, as shown in Table 9. Sample visual observation shows a darker color for the UA castable in comparison to the NA castable. The maximal penetration depth of 15.24 mm was measured for the UA castable, whereas the NA castable performed better results and the value of 14.87 mm was obtained. In addition, the total corroded area A was higher by 3.9% for the UA castable than the NA castable, although the apparent porosity of the NA castable is higher than the UA castable (+0.2% after firing at 1500 °C).

Corroded crucibles were also examined using SEM analysis, element mapping and phase analysis. Cross-cut sections of corroded crucibles were analyzed using SEM and its outputs are presented in Appendix A—Figure A1, and Appendix B—Figure A3. Corroded and non-corroded areas were sectioned-cut under water, samples were then dried, their surface was goldened and the microstructure was observed using SEM. Comparing the microstructure of the corroded and non-corroded samples, the corroded sample shows a higher amount of glassy phase and newly created minerals caused by corrosion. This finding was also confirmed by quantitative phase analysis (Appendix A—Figure A2 for NA, Appendix B—Figure A4 for UA), where an increase in the glassy phase of 4.4% for NA was measured and 2.7% for UA, respectively. The new mineral detected in the phase composition of the corroded samples is kalsilite K(AlSi_2_O_6_) and its amount is 0.9% for NA and 0.7% for UA. During the corrosion process, the content of mullite decreased by 5.6% for NA and 0.5% for UA. Despite these findings, the no-cement castable resisted corrosion better, as shown in Table 8.

Samples for EDS element mapping were prepared by cutting 20 × 20 mm cubic samples, subsequently polished under water. Outputs are presented in Appendix A—Figure A2, and in Appendix B—Figure A4. The EDS probe localized the presence of the K elements in both sections—the optically visual transition zone is indicated at a magnification of 100× by the grey dashed line. When analyzing the NA castable, the peak intensity for the K element was measured two times higher for the corroded section in comparison to the non-corroded section. Analysis of the UA castable shows a four-times-higher peak intensity for the K element in the corroded section than in the non-corroded section. When comparing the bond type, the UA castable visually appears to be more infiltrated by K_2_CO_3_ based on the presence of K^+^ ions. This is caused by a higher average pore size although the apparent porosity is slightly higher for the NA castable. The castable containing pores with a higher diameter allows the corrosive media to infiltrate the castable easier than the castable containing pores with a lower diameter. Additionally, calcium-containing mixtures or castables usually perform with a lower refractoriness than calcium-free mixtures [10].

## 4. Conclusions

Fine pastes prepared from the fine matrix and five types of colloidal silica performed comparable properties with minimal differences (especially for BD and CMOR). Higher differences were obtained in AP. To achieve a high density paste, K1530K colloidal silica is the most suitable.The phase composition of fine pastes after firing can be slightly modified using a different type of colloidal silica. Increasing the content of SiO2 solids by 10% in colloidal silica results in a 1.3% increase of the mullite content (at 1500 °C).The performance and properties of castables can be controlled by using different types of grog. The most stable properties after firing were obtained using fireclay and mullite grog. To achieve specific properties at specific conditions, the castables mixture may be tailored for its purpose—e.g., using 70% of A and 30% of M to achieve better abrasion resistance and a permanent linear change close to the zero.The pore structure mainly influences the corrosion resistance. A higher average pore diameter leads to deteriorating the corrosion resistance. Mercury intrusion porosimetry (MIP) can be used to evaluate the proper castable design in terms of the pore structure, especially for castables used in corrosive environments.The sol-gel-bonded castable, compared to the more traditional calcium aluminate-bonded castable, performed successfully at all tested aspects. Key property values such as permanent linear changes, the apparent porosity and bulk density were almost at the same level regardless of the used bond. The sol-gel-bonded castable performed better for corrosion resistance.

## Figures and Tables

**Figure 1 materials-15-05918-f001:**
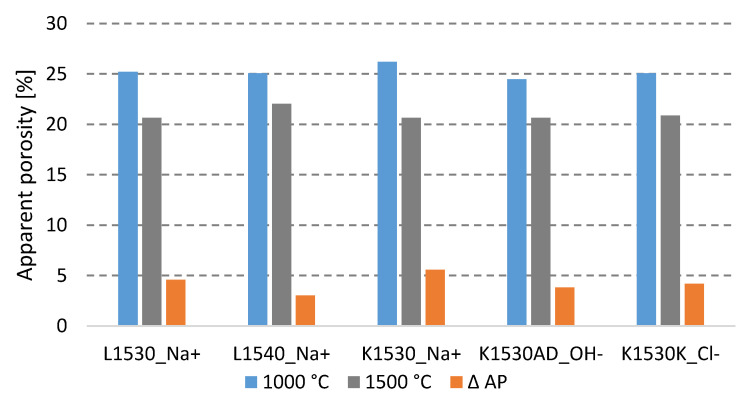
Comparison of different colloidal silica and its influence on apparent porosity after firing at 1000 °C and 1500 °C.

**Figure 2 materials-15-05918-f002:**
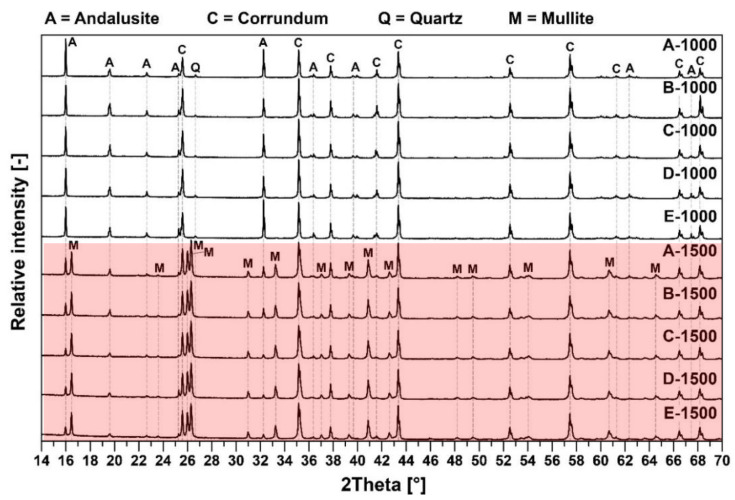
Mineralogical composition of the fine matrix fired at 1000 °C (white background) and 1500 °C (red background).

**Figure 3 materials-15-05918-f003:**
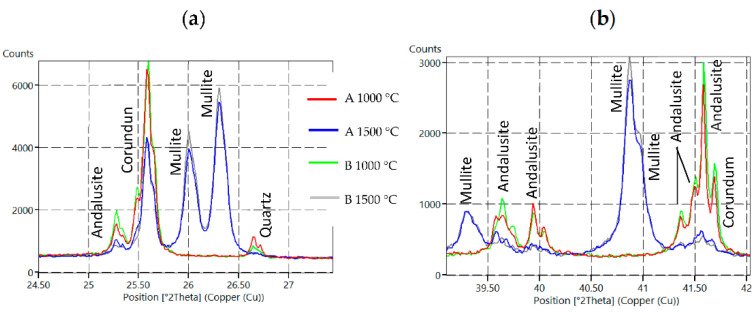
XRD diffractograms for Sample A and B fired at 1000 °C and 1500 °C with focus on specific sections—(**a**) 24.5°–27.5° 2θ section, (**b**) 39.0°–27.5° 2θ section.

**Figure 4 materials-15-05918-f004:**
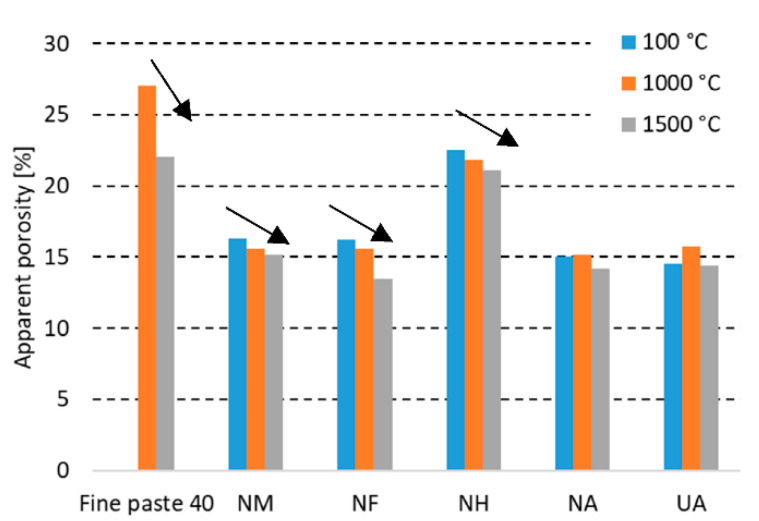
Apparent porosity of tested castables at elevated temperatures.

**Figure 5 materials-15-05918-f005:**
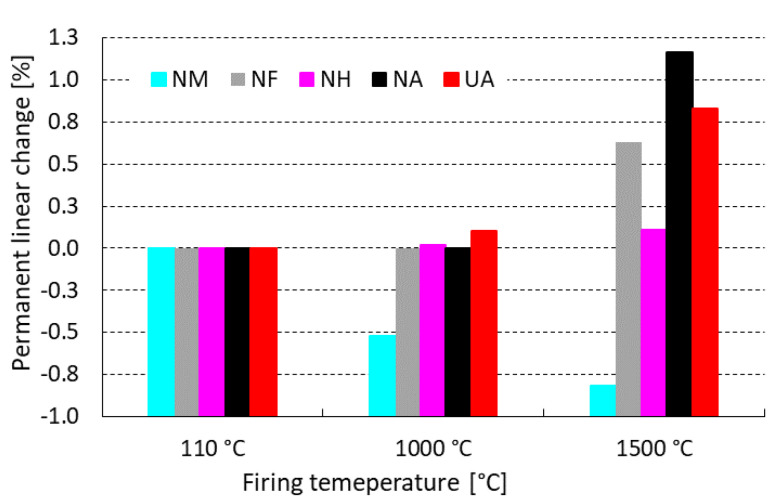
Influence of firing temperature on LCf of tested castables.

**Figure 6 materials-15-05918-f006:**
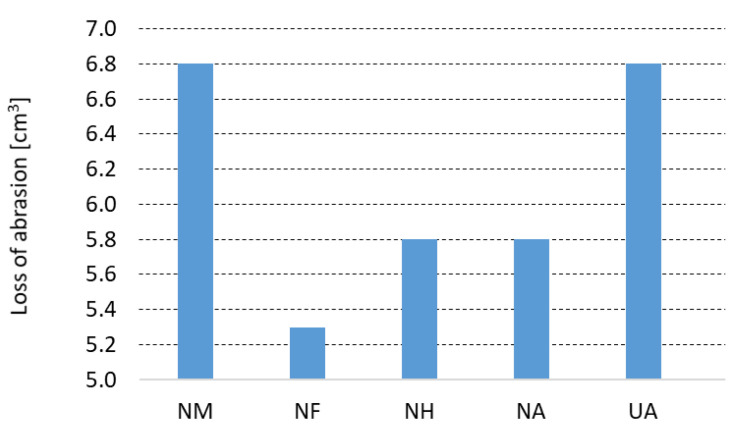
Loss of abrasion determined according to standard ASTM C 704.

**Figure 7 materials-15-05918-f007:**
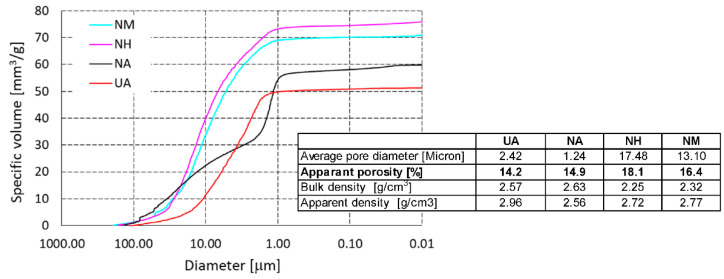
Total cumulative pore volume of laboratory-prepared samples and reference material.

**Table 1 materials-15-05918-t001:** Chemical and phase composition of fine matrix compounds in wt. %.

Compound/ Composition	Fine Matrix Compounds				
	A	TA	RA	GA	SF
Chemical composition					
SiO_2_	37.01	0.09	0.16	0.80	98.46
Al_2_O_3_	60.49	99.55	99.45	98.35	0.04
TiO_2_	0.19	–	–	0.03	0.01
Fe_2_O_3_	0.95	0.01	–	0.09	0.07
CaO	0.20	–	0.01	0.22	0.62
MgO	0.18	–	0.14	0.23	0.04
K_2_O	0.29	–	0.04	0.03	0.55
Na_2_O	0.29	0.45	0.20	0.25	0.01
Phase composition					
Quartz	1.37	–	–	–	0.26
Corundum	–	94.98	100	98.90	–
Andalusite	90.12	–	–	–	–
Diaoyudaoite	–	5.02	–	1.10	–
Amorphous phase	8.51	–	–	–	99.74

**Table 2 materials-15-05918-t002:** Composition of fine matrix in % wt.

Component	Proportion in the Fine Matrix [% wt.]
Andalusite <400 µm	16.75
Tabular alumina <44 µm	18.25
Reactive alumina 0.5–3.0 µm	36.00
Ground alumina 0.5–3.0 µm	14.75
Silica fume <0.15 µm	14.25

**Table 3 materials-15-05918-t003:** Chemical and phase composition of coarse grog in wt.%.

Compound/ Composition	Coarse Grog			
	M	F	H	A
Chemical composition				
SiO_2_	22.92	56.48	47.38	36.52
Al_2_O_3_	76.57	39.66	50.33	59.74
TiO_2_	0.00	1.40	0.97	0.24
Fe_2_O_3_	0.05	1.00	0.50	0.58
CaO	0.14	0.50	0.07	0.26
MgO	0.02	0.11	0.13	0.11
K_2_O	0.10	0.30	0.30	0.05
Na_2_O	0.20	0.55	0.13	2.43
Phase composition	M	F	H	A
Quartz	0.1	2.5	0.1	1.4
Mullite	80.5	35.7	78.6	–
Corundum	0.2	–	1.5	–
Cristobalite	–	7.4	2.1	–
Andalusite	–	–	–	90.1
Amorphous phase	19.2	54.4	17.7	8.5

**Table 4 materials-15-05918-t004:** Designed composition of castables in % wt.

Designation	NM	NF	NH	NA	UA
Component	[%]	[%]	[%]	[%]	[%]
Castable type	NCC	NCC	NCC	NCC	ULCC
Coarse grog	Mullite	Fireclay	High alumina	Andalusite	Andalusite
Coarse grog content	78.0	78.0	78.0	78.0	78.0
Fine matrix	22.0	22.0	22.0	22.0	22.0
Water	2.2	5.5	3.6	3.1	5.5
L1540	4.75	4.75	4.75	4.75	-
CAC	-	-	-	-	2.0

Legend: N—no cement castable, U—ultra-low cement castable.

**Table 5 materials-15-05918-t005:** Five types of colloidal silica and their influence on final properties of fine pastes after firing [% wt.].

Sample	Colloidal Silica	Stabilization	T [°C]	BD [kg∙m^−3^]	AP [%]	CMOR [MPa]	LCf [%]
A	L1530	Na^+^	1000	2490	25.21	6.2	−0.2
			1500	2599	20.54	6.4	−1.9
B	L1540	Na^+^	1000	2510	25.07	6.2	−0.1
			1500	2565	22.04	6.2	−1.0
C	K1530	Na^+^	1000	2493	26.21	6.1	−0.4
			1500	2590	20.66	6.4	−2.5
D	K1530AD	OH^−^	1000	2517	24.46	6.6	−0.3
			1500	2608	20.74	6.6	−2.2
E	K1530K	Cl^−^	1000	2512	25.08	6.1	−0.2
			1500	2600	20.88	6.6	−1.9

T—temperature, BD—bulk density, AP—apparent porosity, CMOR—cold modulus of rupture, LCf—linear change after firing (1000 °C).

**Table 6 materials-15-05918-t006:** Phase composition of samples fired at 1000 °C and 1500 °C [%].

Firing Temperature [°C]	1000 °C					1500 °C				
Samples Formula	A	B	C	D	E	A	B	C	D	E
Quartz	0.5	0.6	0.6	0.5	0.5	0.2	0.1	0.2	-	0.1
Corundum	68.8	67.9	67.0	66.7	67.9	28.7	28.6	28.9	28.9	30.1
Andalusite	15.1	15.7	16.3	16.2	15.1	-	-	-	-	-
Mullite	-	-	-	-	-	71.1	71.3	70.9	71.1	69.8
Amorphous phase	15.6	15.8	16.1	16.7	16.7	-	-	-	-	-

**Table 7 materials-15-05918-t007:** Calculated chemical compositions of castables based on WDXRF analysis in % wt.

Mixture	SiO_2_	Al_2_O_3_	TiO_2_	Fe_2_O_3_	CaO	MgO	K_2_O	Na_2_O
NM	24.27	74.91	0.05	0.25	0.03	0.08	0.03	0.18
NF	42.13	54.79	0.95	0.79	0.13	0.30	0.54	0.17
NH	39.45	58.49	0.82	0.56	0.06	0.08	0.20	0.15
NA	33.05	65.49	0.11	0.68	0.08	0.12	0.11	0.15
UA	32.32	65.45	0.16	0.88	0.79	0.04	0.01	0.13

**Table 8 materials-15-05918-t008:** Physical properties of tested castables after drying at 110 °C and after firing at 1000 °C and 1500 °C.

Mixture	T [°C]	BD [kg∙m^−3^]	WA [%]	AP [%]	LCf [%]
NM	110	2690	6.1	16.3	-
	1000	2700	5.8	15.6	−0.52
	1500	2700	5.7	15.2	−0.82
NF	110	2410	6.7	16.2	-
	1000	2430	6.4	15.6	0.00
	1500	2370	5.7	13.5	0.63
NH	110	2300	9.8	22.5	-
	1000	2320	9.4	21.8	0.02
	1500	2315	9.1	21.1	0.11
NA	110	2730	5.5	15.0	-
	1000	2715	5.6	15.2	0.00
	1500	2630	5.4	14.2	1.16
UA	110	2740	5.3	14.5	-
	1000	2700	5.8	15.7	0.10
	1500	2640	5.5	14.4	0.83

Legend: T—temperature, BD—bulk density, WA—water absorption, AP—apparent porosity, LCf—linear change after firing.

**Table 9 materials-15-05918-t009:** Visual evaluation of crucible test for tested castables.

Castable	UA	NA
Corrosion category	U	U
d_max_ [mm]	15.24	14.87
A [mm^2^]	1376	1326 (−3.9%)

## Data Availability

The data presented in this paper are available upon request from the corresponding author.

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
