# Peer review of "Development and Testing of Castables with Low Content of Calcium Oxide"

_materials, 2022, doi:10.3390/ma15175918_

Round 1
Reviewer 1 Report
The publication is written correctly from the methodological side. The work contains elements of scientific novelty. The literature is correctly selected.
I only have a small editorial note,
a) for Figure 5, I suggest using a bar graph similar to Figure 4
b) section 2 Methods should include Matherials and Method. This section should be done more precisely, describing separately all the research techniques used, describing the apparatus, materials and procedures used.
Author Response
Dear reviewer,
thank you very much for your review. Your points were noted and incorporated to the manuscript.
Please see the attechment where we responded the points separately.
Sincerely
Authors (David Zemánek and Lenka NevÅ™ivová)

Reviewer 2 Report
The use of colloidal systems such as the colloidal silica in this work is an important alternative for hydraulic bonds (cement, hydratable alumina) to increase the purity of refractory material and hence increase its thermomechanical properties. The work shows novelties and the presentation is of proper quality. However, there is a major concern about the work:
Generally, the focus on colloidal silica bonding vs. cement bonding is too narrowly. The main concern of calcia in a fire clay mixture comes with the application temperature. But you don’t mention any application (temperature). Furthermore, you compare castables sintered at 1000 °C and 1500 °C and choose a ULCC as a reference. This is correct for 1500 °C, but not for 1000 °C. If you focus on application at 1000 °C (e.g. aluminium melting), there is no need to abstain from high calcia cement. Hence, an MCC or LCC would be a much better reference and will at 1500 °C support your approach, where your colloidal silica batches will probably show better high-temperature properties. And next to cement bonding, there are fire clays with e,g, clay bonding or kaolin bonding. If you are focusing on high-temperature properties, why not choose hydratable alumina?! This will also increase the amount of mullite and decrease the amount of amorphous phase you have in your mixtures. Please add a discussion about these concerns to your manuscript.
Additionally, other points need minor revision before publication:
1. Please check your references again. For instance, I recognized that you referred to [9] when presenting CMA cement (page 2, line 60), but CMA was introduced by C. Parr and C. Wöhrmeyer in 2011 in the proceedings of the UNITECR. So please cite the primary source. Your reference [9] is from 2004 and hence cannot include CMA.
2. Page 4, line 158; page 5, line 192; page 9, line 296, and section 3.6: the unit of heating rate is K/min.
3. Section 3.2. Please explain why you prepared a paste containing all the fine matrix compounds A, TA, RA, GA, and SF instead of choosing one matrix compound and mixing it with each of the 5 colloidal silica. And please explain the chosen proportions in table 2.
4. Section 3.6: You stated, that the apparent porosity of NA castable is higher than UA castable after sintering at 1000 °C. On the one hand, a corrosion test was performed on samples after sintering at 1500 °C. Hence, you should compare the porosity after sintering at 1500 °C (maybe “1000 °C“ is a writing mistake). On the other hand, the pore size and pore size distribution has also a major effect on the infiltration. Please revise this part and add a statement about the pore size, next to AP (page 12, lines 382-384).
5. Table 8: The evaluation of the corrosion test is wrong. Both crucibles are of corrosion category U – unaffected. There is just an infiltration but no degradation visible. Furthermore, it is still not clear why you choose K2CO3 as the attacking medium. What is the proposed application of your new formulations? Again, keep in mind my general concern about your work: if you are focusing on corrosion at 950 °C, cement bonding would be an excellent choice! At 950 °C, there will be no loss of thermomechanical properties due to the CaO. Furthermore, keep also in mind that there is a decomposition of K2CO3 and you will have evaporation of potassium. And at this temperature, corrosion is not the first degradation mechanism, but spalling. Therefore, a disc test or gradient furnace test was developed, due to the expansion of the microstructure by interaction with K, not corrosion (dissolution).
Author Response
Dear reviewer,
thank you very much for your fruitful review. Your points were noted and incorporated to the manuscript.
Please see the attechment where we responded the points separately.
Sincerely
Authors (David Zemánek and Lenka NevÅ™ivová)
